# Heterogeneity Challenges in Multiple-Element-Modified Lead-Free Piezoelectric Ceramics

**DOI:** 10.3390/ma12244049

**Published:** 2019-12-05

**Authors:** Oana Andreea Condurache, Kristian Radan, Uroš Prah, Mojca Otoničar, Brigita Kmet, Gregor Kapun, Goran Dražić, Barbara Malič, Andreja Benčan

**Affiliations:** 1Electronic Ceramics Department, Jožef Stefan Institute, Jamova 39, 1000 Ljubljana, Slovenia; kristian.radan@ijs.si (K.R.); uros.prah@ijs.si (U.P.); mojca.otonicar@ijs.si (M.O.); brigita.kmet@ijs.si (B.K.); goran.drazic@ijs.si (G.D.); barbara.malic@ijs.si (B.M.); andreja.bencan@ijs.si (A.B.); 2Jožef Stefan International Postgraduate School, Jamova 39, 1000 Ljubljana, Slovenia; 3National Institute of Chemistry, Hajdrihova 19, 1000 Ljubljana, Slovenia; gregor.kapun@ki.si

**Keywords:** lead-free, piezoceramics, KNN, heterogeneity, TEM

## Abstract

We report on a heterogeneity study, down to the atomic scale, on a representative multiple-element-modified ceramic based on potassium sodium niobate (KNN): 0.95(Na_0.49_K_0.49_Li_0.02_)(Nb_0.8_Ta_0.2_)O_3_–0.05CaZrO_3_ with 2 wt % MnO_2_. We show that different routes for incorporating the MnO_2_ (either before or after the calcination step) affect the phase composition and finally the functionality of the material. According to X-ray diffraction and scanning electron microscopy analyses, the ceramics consist of orthorhombic and tetragonal perovskite phases together with a small amount of Mn-rich secondary phase. The addition of MnO_2_ after the calcination results in better piezoelectric properties, corresponding to a ratio between the orthorhombic and tetragonal perovskite phases that is closer to unity. We also show, using microscopy techniques combined with analytical tools, that Zr-rich, Ta-rich and Mn-rich segregations are present on the nano and atomic levels. With this multi-scale analysis approach, we demonstrate that the functional properties are sensitive to minor modifications in the synthesis route, and consequently to different material properties on all scales. We believe that detecting and learning how to control these modifications will be a step forward in overcoming the irreproducibility problems with KNN-based materials.

## 1. Introduction

As European legislation limits the use of lead-based piezoelectrics [1], there are high expectations for materials based on potassium sodium niobate (KNN). The breakthrough was made by Saito et al. in 2004, who reported an improved piezoelectric coefficient for Ta- and Li-modified KNN, comparable with that of lead-based materials [2]. It was envisaged, for the first time, that a modified KNN could overcome the problem of the modest piezoelectric activity of pure KNN ((K_0.5_Na_0.5_)NbO_3_), triggering the development of a lead-free piezoelectric market.

Multiple-element modifications of KNN are used to increase its functionality. On the one hand, the piezoelectric activity is increased by introducing phase coexistence at room temperature. For example, doping with elements like Sr, Ta, Zr, Ti, Bi, Ba and Ca has been shown to decrease the orthorhombic–tetragonal transition temperature down to ambient temperatures. On the other hand, Li is frequently added to compensate for the reduction of the Curie temperature induced by the other dopants [3]. Moreover, since densification is challenging in KNN [4], the synthesis usually involves sintering aids such as MnO_2_ [5], ZnO [6], (K,Na)-germanate [7] or Cu-based materials [8]. 

However, when multi-element modifications are involved, the risk of chemical inhomogeneity is high. Therefore, the problem of the reproducibility and reliability of KNN-based materials comes to the fore, as already acknowledged by a few authors [3,9,10]. For instance, 0.95(Na_0.49_K_0.49_Li_0.02_)(Nb_0.8_Ta_0.2_)O_3_–0.05CaZrO_3_ (KNLNT–CZ) with 2 wt % MnO_2_ has excellent piezoelectric properties combined with a high temperature stability. Nonetheless, different piezoelectric coefficients have been reported for this composition [11,12], which suggests that the distribution of element species, their possible segregation or inability to incorporate into the structure, and the formation of secondary phases, are all likely to have a major impact on the piezoelectric performance. Studies of the chemical inhomogeneity and secondary phases can be found for single- or a-few-elements-doped KNN [13,14,15,16], while for multiple-elements-modified systems there is a lack of literature data, especially down to the nano and atomic levels. 

In the present study, we investigated two ceramics with the KNLNT–CZ composition, for which MnO_2_ was added either (i) before, or (ii) after the calcination step. MnO_2_ does not only help densification, but it was also shown to have an important role in the piezoelectric properties. For example, in Li-modified KNN the addition of 1.2 mol % MnO_2_ was reported to cause a four-fold increase in the mechanical quality factor *Qm* and a decrease of the *d_33_* piezoelectric coefficient by 30% [17]. In contrast, the *d_33_* in KNLNT-CZ increased by a factor of 2 when adding 2 wt % MnO_2_ [18]. Some authors reported adding MnO_2_ before the calcination together with other precursors [19], but there are other reports of adding it after the calcination process [17,20]. It is not clear what influence the sequence of synthesis steps (i.e., when the MnO_2_ is added) has on the phase composition, the formation of secondary phases and, finally, on the functional performance of the material. 

We have attempted to correlate the structure and the chemical composition to the functional properties of KNLNT-CZ-MnO_2_ ceramics. Using a multiscale approach, by combining electron microscopy together with analytical tools, detailed micro- to atomic-scale analyses were performed. The main aim of the study was to prove that the functional properties are, to a large degree, sensitive to minor modifications of the synthesis route, and consequently to different material properties on all scales. Furthermore, we show that when many dopants are involved, even in samples that are apparently homogeneous at the macroscopic level, the possibility of element/phase segregation and defects at the nano and atomic levels is very high.

## 2. Materials and Methods 

Ceramics with the nominal composition 0.95(Na_0.49_K_0.49_Li_0.02_)(Nb_0.8_Ta_0.2_)O_3_–0.05CaZrO_3_ with 2 wt % MnO_2_ were prepared using a mechanochemically assisted synthesis route. The precursors involved in the synthesis were: K_2_CO_3_ (99.9%, ChemPur, Karlsruhe, Germany), Na_2_CO_3_ (99.9%, ChemPur, Karlsruhe, Germany), Li_2_CO_3_ (98.5%, Riedel-de Haën AG (RdH), Seelze-Hannove, Charlotte, Germany), Nb_2_O_5_ (99.9%, Sigma-Aldrich, St. Louis, MI, USA), Ta_2_O_5_ (99.85%, Alfa Aesar, Ward Hill, MA, USA), CaCO_3_ (99.95%, Alfa Aesar, Ward Hill, MA, USA), ZrO_2_ (99.1%, Tosoh, Tokyo, Japan) and MnO_2_ (99.9%, Alfa Aesar, Ward Hill, MA, USA). Two different samples were prepared: one for which MnO_2_ was added before the calcination step (denoted as Mn-BC) and one sample for which the MnO_2_ was added after the calcination step (denoted as Mn-AC). The homogenized stoichiometric mixtures, i.e., one with and one without MnO_2_, were mechanochemically activated by high-energy milling in a planetary mill (Retsch, Model PM 400, Haan, Germany) for 8 h at a disk rotational frequency of 300 min^−1^ and a vial-to-disk rotational frequency of −3, using a tungsten carbide milling vial (volume 80 cm^3^) filled with WC milling balls (ball diameters 10 mm). Further, the activated powder was pressed into pellets and calcined at 800 °C for 4 h. The calcined pellets were then crushed and the resulting powder was milled. Finally, in the case of the Mn-BC sample, the calcined powder was cold isostatically pressed under 200 MPa into pellets and sintered in air at 1150 °C for 2 h with a heating/cooling rate of 5 °C·min^−1^. For the other sample, Mn-AC, MnO_2_ was added to the calcined powder and homogenized, then the mixture was cold isostatically pressed under 200 MPa into pellets and sintered in air at 1110 °C for 2 h with a heating/cooling rate of 5 °C/min. The absolute densities of the sintered pellets were determined with a helium pycnometer (AccuPyc II 1340, Micromeritics, Norcross, GA, USA). A more detailed description of the preparation method can be found elsewhere [12]. 

To characterize the samples’ electrical properties, the ceramic pellets were cut, ground down to 0.35 mm in height, polished and then Cr/Au electrodes were sputtered using a RF-magnetron machine (5Pascal, Milano, Italy). Permittivity (*ε*) and dielectric losses (*tan δ*) were measured with a Hewlett Packard 4192A Impedance Analyzer (Hewlett Packard, Palo Alto, CA, USA). The coupling coefficient *k_p_* was obtained using the resonance method and the *d_33_* coefficient was determined with a Berlincourt piezometer (Take Control PM10, Birmingham, UK). Prior to these measurements the samples were poled with a 4 kV·mm^−1^ bias at 120 °C in a silicone oil bath for 40 min and field-cooled.

The X-ray diffraction (XRD) analyses were performed on the crushed sintered pellets with a PANalytical X’Pert PRO (PANalytical, Almelo, Netherlands) high-resolution diffractometer using Cu-*Kα1* radiation (λ = 1.54056 Å) equipped with a 100-channel X’Celerator detector. Diffraction patterns were collected at room temperature over a 2*θ* range from 10 to 90° with a step of 0.017° and an integration time of 200 s per step.

The microstructural analyses of the ceramics were performed using a Jeol JSM-7600F field-emission scanning electron microscope (SEM, Jeol, Tokyo, Japan) equipped with an INCA Oxford 350 EDS SDD energy-dispersive X-ray spectroscopy system (EDXS, Oxford Instruments, Abingdon, UK). Ceramic specimens were prepared for SEM analysis using standard metallographic methods, and then carbon coated using a PECS 682 (Gatan, Pleasanton, CA, USA). Grain size analyses were performed on thermally etched samples, from the backscattered-electron SEM (BSE-SEM) images based on Feret’s statistical diameter measurement [21] using at least 500 grains and Image Tool software (v3.0, developed by Wilcox et al., University of Texas Health Science Center, Austin, TX, USA).

Structural and chemical investigations on the nano and atomic scales were performed with transmission (TEM) and scanning transmission electron microscopy (STEM). JEM-2100 and JEM-2010F (Jeol Ltd., Tokyo, Japan), equipped with a LINK ISIS 300 EDXS system (Oxford Instruments, Oxfordshire, UK) and Jeol ARM 200CF (Jeol Ltd., Tokyo, Japan), equipped with a Jeol Centurio 100 mm^2^ SDD EDXS system and a Gatan Quantum ER Dual Electron Energy Loss Spectroscopy (EELS) system (Gatan, Pleasanton, CA, USA), were utilized. Specimens were prepared in two ways: i) by grinding, dimpling and final Ar milling (Gatan PIPS Model 691, New York, NY, USA) and ii) by a focused ion-beam (FEI Nanolab Helios 650-FIB, Waltham, MA, USA) lamella preparation procedure. Selected-area electron diffractions (SAED) were simulated using the CrystalMaker software (v2.2, CrystalMaker Software Ltd., Oxfordshire, UK) [22].

## 3. Results and Discussions

### 3.1. Macroscopic Properties

The functional properties and absolute densities of Mn-BC and Mn-AC are given in Table 1. Adding the MnO_2_ after the calcination was clearly favorable for improving the piezoelectric activity in terms of the longitudinal piezoelectric coefficient *d_33_* (238 pC·N^−1^ for Mn-AC compared with 140 pC·N^−1^ for Mn-BC) and the electromechanical coupling coefficient *k_p_* (40% compared with 27%, respectively). The dielectric properties were comparable, with slightly higher dielectric losses for the Mn-AC (0.050 compared with 0.034). The absolute densities were similar for both ceramics: 4.86 g·cm^−3^ for Mn-BC and 4.65 g·cm^−3^ for Mn-AC.

The phase investigation by XRD is shown in Figure 1, while the phase composition, determined by Rietveld refinement, is given in Table 2. Both samples are a two-phase composition according to the best refinement fit (See Appendix A), including orthorhombic (O) and tetragonal (T) symmetries of the KNLNT perovskite, together with a small amount of tetragonal Mn_3_O_4_ secondary phase (see Figure 1a). The peaks assigned to the secondary phase had a low intensity; some of the peaks are magnified in Figure 1b. In Figure 1c, the fitting results of the (310) peak show the co-existence of orthorhombic (*Bmm*2) and tetragonal (*P*4*mm*) symmetries for both KNLNT ceramics.

We note that 2 wt % MnO_2_ is a hyper-stoichiometric quantity [12], so a secondary phase of Mn_3_O_4_, due to the thermal evolution of the initial MnO_2_ (i.e., MnO_2_→Mn_2_O_3_→Mn_3_O_4_) [23], is expected. According to the XRD, the quantity of Mn_3_O_4_ secondary phase, which starts forming at T 750 °C [23], is comparable in the Mn-BC and Mn-AC ceramics. 

From the point of view of the main perovskite phase’s composition, the two analyzed ceramics show different amounts of the orthorhombic and tetragonal perovskite phases (Table 2). Adding MnO_2_ before the calcination step (sample Mn-BC) promotes the formation of the orthorhombic perovskite phase (O/T of 1.9), while, in contrast, when added after the calcination step (sample Mn-AC), the O/T is closer to unity, with slightly more content of tetragonal phase (O/T of 0.7). The coexistence of the tetragonal and orthorhombic phases in KNLNT-CZ has been reported by a few authors [11,12,24,25]. It was suggested that a higher piezoelectric response could be obtained for a ratio of orthorhombic to tetragonal that is closer to unity [24], which is consistent with our findings. What is more, it was reported that in KNN-based materials, where the tetragonal and orthorhombic phases coexist, the increasing amount of Mn promotes the formation of the orthorhombic perovskite phase [17,26]. Based on this, the present study implies that the different incorporation approach of the MnO_2_ can lead to different amounts of Mn entering the perovskite lattice, as well as to the creation of different point defects, as was previously shown for KNLNT-CZ [18] and BiFeO_3_-BaTiO_3_ [27]. This has a substantial effect on the final phase composition and functional properties_._

### 3.2. Microstructure and Chemical Composition (from the Micro to the Atomic Level)

To better understand the differences between the two samples, we went further by analyzing the microstructure of the ceramics, as well as the chemistry on the local scale. 

Micrographs of the polished ceramics are presented in Figure 2; the Mn-rich secondary phase is marked by arrows, while the small black dots are pores. It can be seen from Figure 2 that Mn-AC had a higher porosity, which agreed with its lower density compared to Mn-BC. The morphology of the secondary phase was different depending on the step at which the MnO_2_ was added. When MnO_2_ was added before the calcination step, two types of Mn-rich phases form, one with idiomorphic grains (with developed crystal faces; marked with the full arrow in Figure 2a), and one with xenomorphic grains (taking the shape of the surrounding matrix grains, marked with the open arrow in Figure 2a). In contrast, when MnO_2_ was added after the calcination step, the Mn-rich phase just had the idiomorphic–type grains (Figure 2b). Regardless of the different morphology of the Mn-rich phases, only Mn and O were detected by the SEM-EDXS point analyses. Consequently, based on the XRD analysis in which just the peaks of Mn_3_O_4_ were present and those of Mn_2_O_3_ and MnO_2_ were absent, we assigned the Mn-rich phase to Mn_3_O_4_.

In order to explore the chemical composition at the micro level of the main KNLNT-CZ phase, SEM-EDXS point analysis was performed on different arbitrary points of the polished ceramics. In this case, the SEM-EDXS results represent the average signal from an interaction depth of a few micrometers [29] (corresponding to an accelerating voltage of 15 keV), implying an average composition of around 10 grains. The atomic concentrations of K, Na, Nb, Zr, Ca and Ta are shown in Table 3, together with the relative standard deviations calculated over 15 measurements. Li could not be detected by the EDXS analysis and the Mn content in the perovskite matrix, as well as, possible presence of tungsten contamination coming from milling equipment, was below the detection limit of the technique. Generally, the relative standard deviation is similar or lower than the uncertainty of the standardless EDXS analysis (found to be around ±5%). Exceptions are Ca and Zr, for which the large standard deviation suggests that their distribution in the matrix was not uniform. When comparing the two samples, Mn-AC showed a poorer distribution of Ta compared to Mn-BC, which could be related to the different processing routes. 

The average grain size was very similar for both ceramics (0.33 ± 0.16 μm for Mn-BC and 0.33 ± 0.10 μm for Mn-AC), as shown in the BSE-SEM thermally etched micrographs (Figure 3a,b). The analysis of the grain size distribution is presented in Appendix A. It could also be observed from Figure 3a,b that some grains had a core-shell-like structure; the contrast in the BSE-SEM analysis implied a different chemical composition of the core compared to the shell. This result would be discussed later. The Mn-BC proved to be more homogeneous than the Mn-AC, exhibiting three times fewer core-shell-like grains. The higher homogeneity achieved in the Mn-BC could be either due to the slightly higher sintering temperature, or it could be a direct consequence of the fact that the MnO_2_ was incorporated before the calcination, as Calisir et al. showed for BiFeO_3_-BaTiO_3_ [27].

TEM-EDXS was employed in order to obtain a detailed chemical composition of the grains that had a core-shell structure, as this technique can provide a much better spatial resolution than SEM-EDXS. The results are presented in Figure 4. As can be seen from the TEM micrographs, the core and the shell differed in their morphology, i.e., the core part appeared more irregular, whereas the shell region appeared slightly more ordered, in some cases with lamellar domains. We determined that the core and shell had different B-site cation distributions, i.e., a Ta-poor and Nb-rich core, while the shell had the opposite trend. The predisposition for the grains to form a core-shell structure in which the core is poor in Ta has already been observed by several authors in Ta-modified KNN ceramics [12,30,31] and was ascribed to the low diffusion of Ta and Na into the pre-formed K/Nb-rich particles. Apart from the different distribution of the B-site cations (i.e., Ta and Nb) in the core and shell of both samples, the A-site cations are preferentially distributed as well. For both samples, the shell contained more Na, slightly more Ca and less K than the core, consistent with previous observations [12,30]. In the case of Na, however, the amount detected in the core and shell was below the nominal values. This analysis clearly shows that different diffusion kinetics of the elements made the homogeneity of the multiple-element-modified KNN difficult to achieve. We did not comment in this section on the Zr concentration because the amount detected in these grains was very low, close to the limit of the detection for EDXS.

We further researched the samples’ homogeneity down to the nano and atomic levels by performing STEM-EDXS mapping, which is presented in Figure 5. We found that both samples contain additional secondary phases at the nano level that were not detected by XRD, nor were they visible in the SEM micrographs. As seen from Figure 5a–d, a Zr-rich phase was found in both ceramics and it typically has a round morphology and a size of tens of nanometers. Such inclusions of round ZrO_2_ particles have been previously reported in Pb(Zr,Ti)O_3_-based ceramics [32] and were assigned to the slow kinetics of Zr and its limited diffusion. Thus, the dissolution of ZrO_2_ appears to be difficult, even with the mechanochemical synthesis, resulting in nano-scale, unreacted ZrO_2_ particles and, therefore, in a loss of the desired stoichiometry. For the sample in which the MnO_2_ was added before the calcination step we detected an Mn-rich phase inclusion (Figure 5a,c), which matches the morphology of the scattered secondary phase seen in the SEM image (compare Figure 5c with Figure 2a). A Ta-rich phase was also detected in the sample in which we added MnO_2_ after calcination (Figure 5b,d), which is in agreement with the SEM-EDXS analysis that showed that Ta is less uniformly distributed in the Mn-AC (Table 3).

Furthermore, an analysis of the grain boundaries was made by STEM-EELS at the atomic level and determined some level of atomic-scale Mn-rich segregation. By comparison, no Mn was detected in the grain. In Figure 6a,b an example of such an Mn-rich grain boundary with the corresponding EELS analysis for Mn-AC is shown (Figure 6c). In Appendix A a similar type of grain boundary for the Mn-BC ceramic is presented, showing a similar situation. Regardless of the route for adding the MnO_2_, the segregation of Mn could be found from the micro down to the atomic level (compare Figure 3, Figure 5 and Figure 6). We could not, however, comment on how much Mn enters the perovskite lattice due to the limitations of the analytical tools used, the uneven distribution/segregation of the elements, or no elements being present in the analyzed areas.

### 3.3. Domain Structure

As the Mn-AC ceramic showed a superior piezoelectric response, we further investigated its domain structure using TEM and SAED analyses. As shown in Figure 7, two types of grains were found; i.e., grains with up to a-few-hundred-nm-large lamellar-type domains (Figure 7a) and grains with irregularly shaped domains (Figure 7d).

Lamellar domains can be explained as tetragonal 90° twins based on the SAED pattern taken along the [100]_pc_ zone axes (pc stands for pseudo-cubic perovskite cell), which shows splitting of the reflection spots (Figure 7b) in the [01–1]_pc_ direction, perpendicular to the domain walls (see arrow in Figure 7a,b). This kind of splitting is obtained due to the polar axis of the neighboring 90° domains being tilted towards each other on account of their c/a ratio; this angle is called the obliquity angle and yields an angle between the two polar axes slightly smaller than 90° [33]. A simulation of 90° twinning in the *P*4*mm* lattice is shown in Figure 7c. In our case the obliquity angle was calculated from the refined cell parameters (Appendix A) to be 0.2°. A lamellar domain configuration was also reported in KNLNT-CZ [34]. 

Besides the grains with well-defined lamellar domains, most of the grains possessed irregularly shaped domains without well-defined domain boundaries. An example of such a grain is shown in Figure 7d. The SAED pattern from this grain exhibited very weak ½ (*eoo*) superstructure reflections in the [100]_pc_ zone (see arrows in Figure 7e). Consequently, we associated the superstructure reflections with the *Bmm*2 orthorhombic symmetry, where forbidden reflections could appear (see arrows in the simulated SAED pattern in Figure 7f), on account of the fact that the reflections were observed to be very weak, and based on the likelihood of the grains with undefined domain structures representing the lower, orthorhombic symmetry. These forbidden reflections could be assigned to some degree of chemical ordering along the basal planes in the structure; however, they could also be a result of double diffraction. The irregular domain morphology in KNN-based materials has been associated with the orthorhombic structure by other authors as well [35].

Therefore, both symmetries found with the XRD were confirmed locally by SAED. Often, grains of alternate orthorhombic and tetragonal domains are reported in modified KNN [36,37]. Thus, it is also highly likely that both symmetries are present within one grain, especially where the segregation of the elements yields a core-shell structured grain. Despite being able to associate within this experiment the different grain morphologies with an average symmetry for each grain, distortions on a smaller scale (a few nanometers) would give an insight into the chemistry-bound local order of these highly complex compounds. Thus, further investigations on the nano and atomic scale are in progress.

## 4. Conclusions

In the present study, using a multi-scale approach, we analyzed the material KNLNT-CZ with 2 wt % MnO_2_. We showed experimentally that the structure and chemistry of the ceramic were sensitive to the incorporation step of the MnO_2_ (either before or after calcination) and that adding it after the calcination was favorable in terms of the piezoelectric functional properties (a *d_33_* of 238 pC·N^−1^ compared 140 pC·N^−1^ and an electromechanical coupling factor of 40% compared with 27%). Furthermore, we found that despite the bulk analyses by XRD and SEM showing a two-phase composition with a minor amount of Mn_3_O_4_ secondary phase, investigations at the nano and atomic levels by TEM and STEM demonstrated several chemical segregations. Ta-, Zr- and Mn-rich phases were found at the level of a few nanometers, together with one atomic layer of Mn-rich phase at the grain boundary. We believed that all the features, defects and segregation from the micro to the atomic level could be the basis for the challenging reproducibility in modified KNN. Special precautions should be taken with the selection of dopants. As an illustration, Zr dissolution into the perovskite matrix was problematic, promoting segregations of ZrO_2_ particles, and this problem has been reported in systems other than KNN [32]. A viable alternative could be replacing CaZrO_3,_ normally used to modify KNN for improving the temperature stability [38], with some other equivalent perovskite systems such as CaTiO_3_ [39]. Likewise, the choice of synthesis route is critical for obtaining homogeneous KNN ceramics. Acknowledging and understanding the challenges associated with the reproducibility in multiple-element-modified KNN will be a step forward in establishing a lead-free piezoelectric material with the potential for market applications.

## Figures and Tables

**Figure 1 materials-12-04049-f001:**
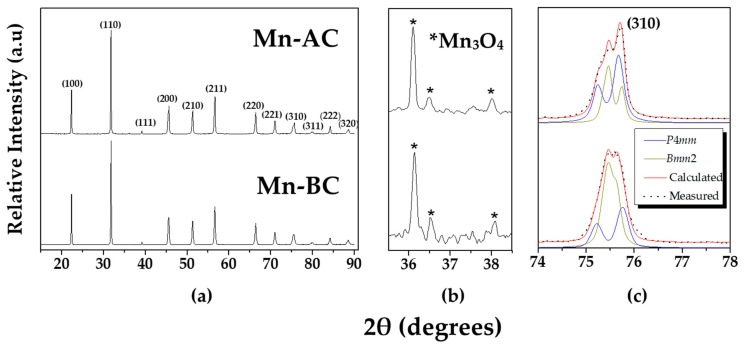
(**a**) XRD patterns of sintered Mn-BC and Mn-AC in the 10–90° 2*θ* range. The diffraction peaks are indexed according to the cubic symmetry (PDF-18-7023, (K_0.47_Na_0.51_Li_0.02_)(Ta_0.1_Nb_0.9_)O_3_ [28]); (**b**) XRD patterns between 35.5 and 38.5° show peaks corresponding to the secondary phase Mn_3_O_4_ (marked by *) and (**c**) the XRD fitting results for the peak (310) show the co-existence of tetragonal (*P*4*mm*) and orthorhombic (*Bmm*2) phases.

**Figure 2 materials-12-04049-f002:**
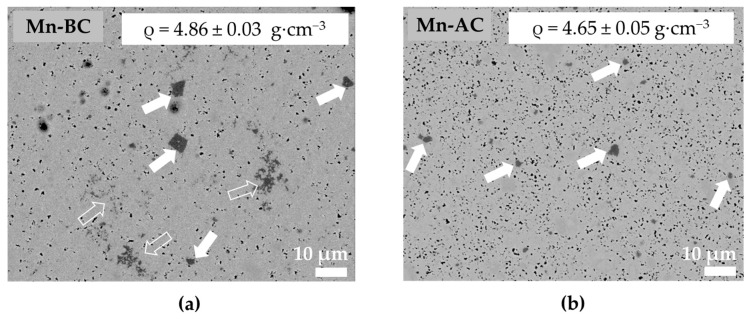
Backscattered-electron SEM (BSE-SEM) micrographs of the polished (**a**) Mn-BC and (**b**) Mn-AC ceramics. The arrows mark the Mn_3_O_4_; a full arrow marks the defined-shaped Mn_3_O_4_ secondary phase, while an open arrow marks the scattered-Mn_3_O_4_ secondary phase. The small black dots are pores.

**Figure 3 materials-12-04049-f003:**
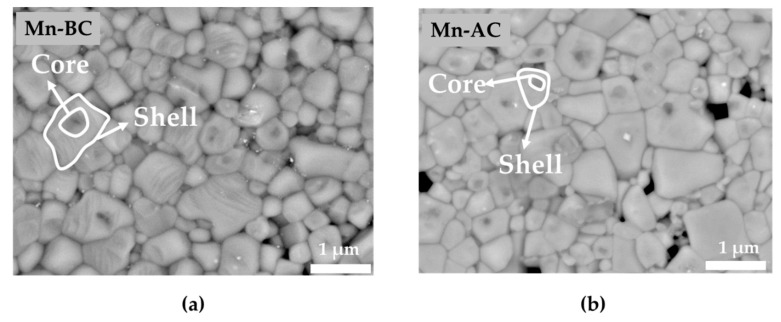
BSE-SEM micrographs of thermally etched (**a**) Mn-BC and (**b**) Mn-AC ceramics; an example of a core-shell-structured grain is marked in each image. The elongated features within the grains in Figure 3a can be atomic planes that etched preferentially or artifacts coming from over thermal etching.

**Figure 4 materials-12-04049-f004:**
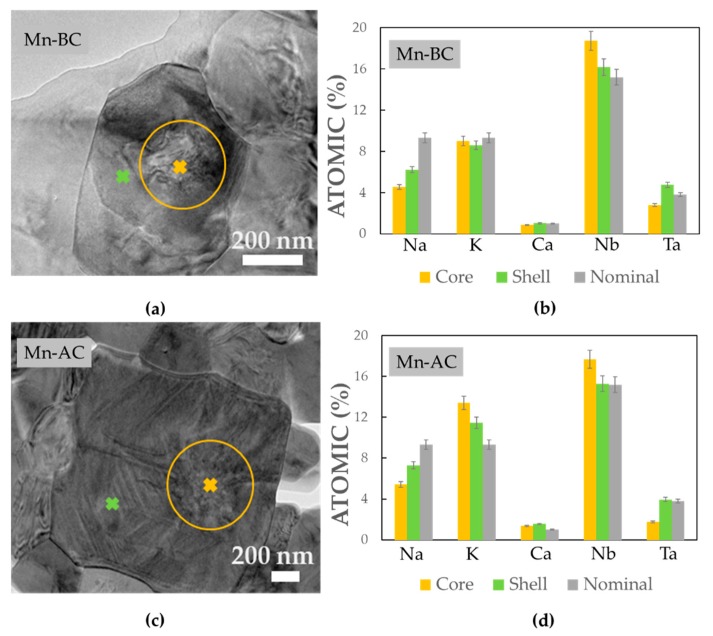
Example of a bright-field TEM (BF-TEM) image of a grain having the core-shell morphology for (**a**) Mn-BC and (**c**) Mn-AC with plots in (**b**) and (**d**) corresponding to the amount of Na, K, Ca, Nb and Ta for the core (orange) and shell (green) areas, together with the nominal composition (grey). The error bar marks a 5% EDXS measurement error. The areas from which the EDXS signal was acquired is marked in (**a**) and (**c**).

**Figure 5 materials-12-04049-f005:**
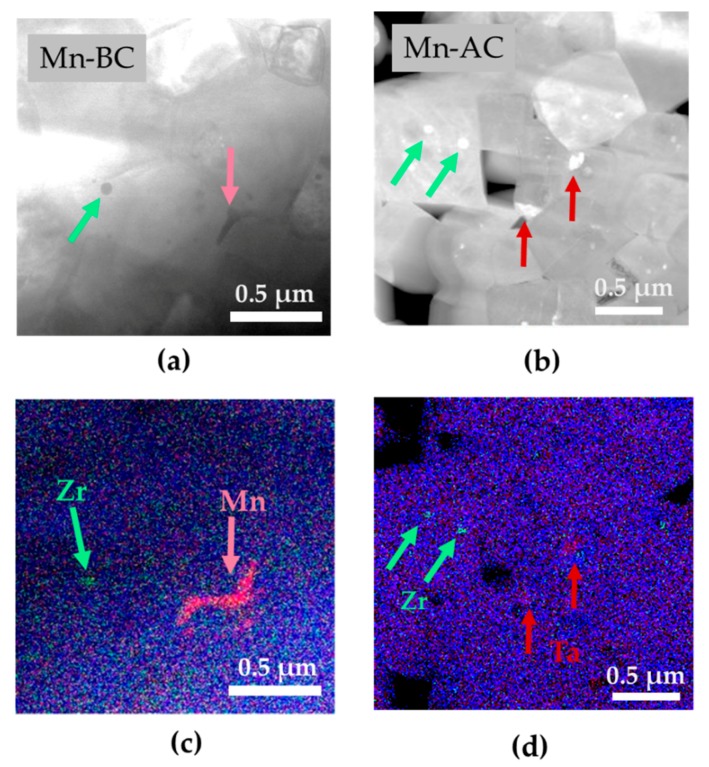
(**a**) Bright-field (BF) STEM image of Mn-BC and (**b**) annular dark-field (ADF) STEM image of Mn-AC with (**c**) and (**d**) corresponding EDXS maps composed of Zr K (green), Mn K (pink), Ta (red) and Nb K (blue) signals.

**Figure 6 materials-12-04049-f006:**
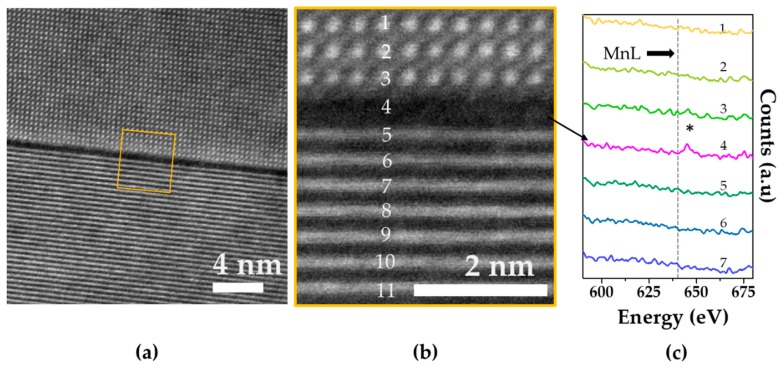
(**a**) ADF STEM image showing a grain boundary in the Mn-AC ceramic; (**b**) enlarged area of the grain boundary with labeled atomic columns/planes where the electron energy loss spectroscopy (EELS) analysis was performed and (**c**) EELS spectra around the Mn L edge from individual atomic planes. Label 4 represents the grain boundary where the Mn L edge (dashed line) was detected (marked with *).

**Figure 7 materials-12-04049-f007:**
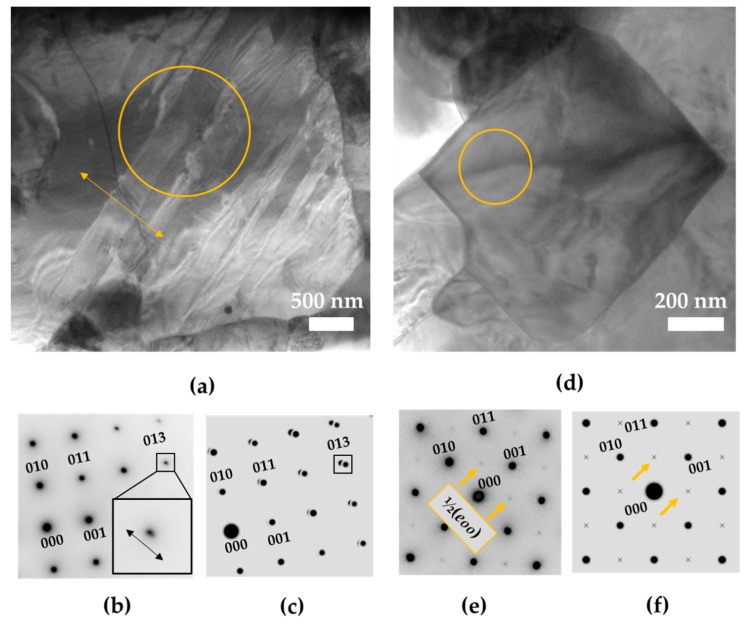
(**a**) BF TEM images of a grain showing lamellar-type domain, together with (**b**) experimental selected-area electron diffractions (SAED) pattern in the [100]pc zone, acquired from the circled area. The insets show the splitting in the [01–1]pc direction of a selected reflection spot, i.e., 013. Double arrow marks the direction of the splitting; (**c**) simulated SAED pattern in the [100]pc zone axis of the 90° tetragonal (*P*4*mm*) domains showing the same type of splitting and (**d**) BF TEM image of a grain showing irregularly shaped domains together with (**e**) experimental SAED pattern in [100]pc zone axis acquired from the circled area. Superstructure reflections of ½ (*eoo*) type are marked with arrows; (**f**) simulated SAED of *Bmm*2 in the [100]pc zone axis. The superstructure reflections are marked with arrows. The indexing of the SAED patterns is made in the pseudo-cubic setting.

**Table 1 materials-12-04049-t001:** Macroscopic properties of Mn-BC and Mn-AC ceramics.

Property	Mn-BC	Mn-AC
Relative permittivity *ε_r_* (room temperature, f = 1 kHz)	1918	2132
Dielectric Losses *tan δ* (room temperature, f = 1 kHz)	0.034	0.050
Piezoelectric coefficient *d_33_* (pC·N^−1^)	140	238
Coupling factor *k_p_* (%)	27	40
Absolute density (g/cm^3^)	4.86 ± 0.03	4.65 ± 0.05

**Table 2 materials-12-04049-t002:** Phase identification by Rietveld XRD refinement.

Sample	KNLNT Orthorhombic (O) (Space Group: *Bmm*2) (wt %)	KNLNT Tetragonal (T) (Space Group: *P*4*mm*) (wt %)	Mn_3_O_4_ Tetragonal (Space Group: *I*4_1_/*amd*) (wt %)	O/T
Mn-BC	65	34	1	1.9
Mn-AC	42	57	1	0.7

**Table 3 materials-12-04049-t003:** Elemental composition by SEM-energy-dispersive X-ray spectroscopy system (EDXS) analysis (in atomic %) for Mn-BC and Mn-AC. Nominal composition is added. The relative standard deviation calculated over 15 measurements is given in parenthesis.

Element	Mn-BC (at %)	Mn-AC (at %)	Nominal (at %)
Na	8.7 ± 0.2 (2.8%)	7.9 ± 0.4 (5.4%)	9.3
K	8.8 ± 0.3 (3.4%)	9.0 ± 0.4 (5.0%)	9.3
Ca	0.9 ± 0.1 (8.2%)	0.9 ± 0.1 (8.6%)	1
Zr	0.8 ± 0.1 (16.4%)	0.8 ± 0.1 (14.3%)	1
Nb	15.7 ± 0.2 (1.4%)	15.8 ± 0.3 (1.7%)	15.2
Ta	4.1 ± 0.1 (2.8%)	4.2 ± 0.3 (7.6%)	3.8

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
