# Peer review of "Heterogeneity Challenges in Multiple-Element-Modified Lead-Free Piezoelectric Ceramics"

_materials, 2019, doi:10.3390/ma12244049_

Round 1

Reviewer 1 Report

  The paper is well organized, but it cannot be accepted in its current version.

English is poor and it is difficult for others to read it. The paper should be edited by a native. In Figure 3, the domain wall is really observed in the Mn-BC ceramics but not in the Mn-AC ceramics, authors should discuss that with the results in Figure 7. How many measured results do you use to calculate the corresponding to the amount of Na, K, Ca, Nb, and Ta?

Reviewer 2 Report

This is a very professional study, including elemental mapping and atomic-resolution imaging. The study involves a small nuanced aspect of the KNN pieoelectric concept, part of a long line of studies on lead-free pieoelectrics,  and therefore can not be considered ground-breaking, but for what it is, it is well done. One issue that needs to be addressed is tungsten contamination. The materials were prepared by 8 hours high-energy milling in a tungsten carbide mill (liner and media). The authors have not investigated the level of tungsten contamination, nor its possible effect on the material. This issue should be addressed.

Round 2

Reviewer 1 Report

Accept in present form